# Adverse Outcome Pathways Associated with the Ingestion of Titanium Dioxide Nanoparticles—A Systematic Review

**DOI:** 10.3390/nano12193275

**Published:** 2022-09-21

**Authors:** Dora Rolo, Ricardo Assunção, Célia Ventura, Paula Alvito, Lídia Gonçalves, Carla Martins, Ana Bettencourt, Peter Jordan, Nádia Vital, Joana Pereira, Fátima Pinto, Paulo Matos, Maria João Silva, Henriqueta Louro

**Affiliations:** 1National Institute of Health Dr. Ricardo Jorge, 1649-016 Lisbon, Portugal; 2ToxOmics—Centre for Toxicogenomics and Human Health, NOVA Medical School, Universidade NOVA de Lisboa, 1169-056 Lisbon, Portugal; 3CESAM, Centre for Environmental and Marine Studies, University of Aveiro, 3810-193 Aveiro, Portugal; 4IUEM, Instituto Universitário Egas Moniz, Egas Moniz-Cooperativa de Ensino Superior, CRL, 2829-511 Monte de Caparica, Portugal; 5Research Institute for Medicines (iMed.ULisboa), Faculty of Pharmacy, University of Lisbon, 1649-003 Lisbon, Portugal; 6NOVA National School of Public Health, Public Health Research Centre, Universidade NOVA de Lisboa, 1600-560 Lisbon, Portugal; 7Comprehensive Health Research Center (CHRC), 1169-056 Lisbon, Portugal; 8BioISI—Biosystems & Integrative Sciences Institute, Faculty of Sciences, University of Lisbon, 1749-016 Lisbon, Portugal; 9NOVA Medical School, Universidade NOVA de Lisboa, 1169-056 Lisbon, Portugal

**Keywords:** titanium dioxide nanoparticles, human exposure, ingested TIO_2_-NPs, AOP, nanosafety, colorectal cancer, adverse outcomes

## Abstract

Titanium dioxide nanoparticles (TiO_2_-NPs) are widely used, and humans are exposed through food (E171), cosmetics (e.g., toothpaste), and pharmaceuticals. The oral and gastrointestinal (GIT) tract are the first contact sites, but it may be systemically distributed. However, a robust adverse outcome pathway (AOP) has not been developed upon GIT exposure to TiO_2_-NPs. The aim of this review was to provide an integrative analysis of the published data on cellular and molecular mechanisms triggered after the ingestion of TiO_2_-NPs, proposing plausible AOPs that may drive policy decisions. A systematic review according to Prisma Methodology was performed in three databases of peer-reviewed literature: Pubmed, Scopus, and Web of Science. A total of 787 records were identified, screened in title/abstract, being 185 used for data extraction. The main endpoints identified were oxidative stress, cytotoxicity/apoptosis/cell death, inflammation, cellular and systemic uptake, genotoxicity, and carcinogenicity. From the results, AOPs were proposed where colorectal cancer, liver injury, reproductive toxicity, cardiac and kidney damage, as well as hematological effects stand out as possible adverse outcomes. The recent transgenerational studies also point to concerns with regard to population effects. Overall, the findings further support a limitation of the use of TiO_2_-NPs in food, announced by the European Food Safety Authority (EFSA).

## 1. Introduction

The technology based on manufactured nanoparticles (NPs) has been pointed as a key enabling technology, due to its potential to improve many products and processes, namely in agriculture, food, and feed industry [1]. Several commercialized products have NPs in its constitution, such as silver, titanium dioxide NPs (TiO_2_-NPs), or synthetic amorphous silica and many others are being developed, such as cellulose nanomaterials. The oral exposure may occur intentionally through the consumption of products containing NPs, or through the ingestion of foods contaminated with NPs released from food-contact materials (packaging, refrigerator coatings, storage containers, or other equipment and coatings) or even through concentration in the food chain [2] due to environmental accumulation. Therefore, along with the dermic and respiratory systems, the gastrointestinal tract (GIT) appears to be a probable route of exposure to NPs that may lead to systemic exposure if the body barriers are surpassed [3]. 

TiO_2_ is one of the NPs most frequently applied in food additives, and in pharmaceuticals and personal hygiene products, such as toothpaste [1], and its consumption (as E171 food additive) is estimated to be 5–9 mg per person per day or even reach up to 32.4 mg/kg per day in children [4]. The exposure by ingestion due to other NPs applied in agriculture, food and feed industry, food packaging, or even food supplements and oral contact products, remains unknown. In 2022, the European Commission (EC) clarified the definition of nanomaterials in a new Recommendation supporting and helping to align legislation across all sectors [5]. The use of TiO_2_ as a food additive was recently considered no longer safe by the European Food Safety Authority (EFSA) [6], and the EC announced the decision to ban its use [5]. The food grade TiO_2_ designated as E171 consists of approximately 40% of TiO_2_-NPs (<100 nm) and 60% of TiO_2_-NPs (>100 nm) [3,7,8]. Nevertheless, other products containing TiO_2_-NPs, such as pharmaceuticals, personal hygiene, or cosmetics, that are not covered in the food regulation, may lead to ingestion of TiO_2_-NPs. For example, in a study from Rompelberg and colleagues [9], it was observed that among Dutch adults, the intake was spread over many food products, food supplements, toothpaste, even in raw cow milk samples, possibly originating from the environment or indirect sources. Whether this exposure may lead to adverse outcomes (AO) has been the subject of research in recent years, but the oral route of exposure remained poorly investigated compared to dermal or inhalation. In spite, it has been suggested that food-grade TiO_2_ may initiate and promote the expansion of preneoplastic lesions in the colon of rats orally exposed [3]. 

Many regulatory agencies across the world, such as the Organization for Economic Cooperation and Development (OECD), or the EFSA, have recognized the potential of Adverse Outcome Pathways (AOPs) in supporting more efficient assessments of chemical safety, as well as for addressing for example biomedical issues or drug development. However, no clear picture has emerged yet between the key events (KE) and the adverse outcomes (AOs) that have been reported upon GIT exposure to NPs, preventing the development of an AOP, as defined by OECD [10]. An AOP is a conceptual framework that organizes previous knowledge concerning biologically plausible and empirically supported links between molecular-level perturbation (named molecular initiating event, MIE) of a biological system and an AO at a level of biological organization of regulatory relevance [10,11]. AOPs development allow to compile the existing information of the biological effects of chemicals in order to present implications for human health and allow decision-making for risk assessors, thereby contributing to protect society from identified adverse health or ecotoxicological effects, such as cancer. Thus, the GIT may represent a target organ for potential adverse effects of ingested NPs. In the present work, TiO_2_-NPs has been selected as a case-study to set up a systematic review for addressing nanosafety concerns that may be applied in the future to other NPs to which the GIT may be exposed. In fact, few studies investigated the behavior of TiO_2_-NPs inside the cells and the molecular pathways triggered that may lead to adverse effects for human health. One major concern for public health is that NPs may produce AOs such as genotoxic effects, that are associated with increased risk of cancer [12]. However, it is also important to characterize other carcinogenic events of a non-genotoxic nature, also triggered by NPs exposure. Although NPs have been extensively investigated in recent years, the studies have generated contradictory results, possibly due to differences in the physicochemical properties of the NPs studied and to other variables in the experimental systems. In a recent publication [13], an updated review about AOPs related with several NPs as stressors, including TiO_2_-NPs, was performed. Consulting the AOP-Wiki, only three published AOPs have been associated with this stressor (AOP 208, 144, and 34), relating it to different AOs such as reproductive failure, steatosis, oedema, and fibrosis in the liver [14,15]. It has been postulated that TiO_2_-NPs may generate ROS and promote oxidative stress and liver inflammation, but it is unknown whether these KEs may cause irreversible AOs in humans. 

The aim of this work is to provide an integrative analysis of the data published on cellular and molecular mechanisms of toxicity related to the ingestion of TiO_2_-NPs, proposing probable KEs that may lead to AOs after exposure, in order to build a comprehensive model for a putative AOP driven by the ingestion of TiO_2_-NPs. 

## 2. Materials and Methods

We performed a literature review, based on Prisma Methodology [16], about the existing data on cellular and molecular mechanisms of toxicity of ingested TiO_2_-NPs in order to establish associated AOPs. The search string considers three main pillars: the subject definition, the route of exposure, and the toxic effect. The search was performed on 12 March 2020, and the search string used was: ((“Titanium dioxide” OR “Titanium dioxide nanoparticle” OR nanotitanium OR “nano titanium” OR “Titanium dioxide nanomaterial” OR “TiO_2_ nanomaterial” OR “TiO_2_ nanoparticles”) AND (gastrointest* OR intestine* OR oral* OR ingest* OR Food OR Pack* OR Water* OR Adsorption) AND (Genotoxic* OR Cancer OR Toxic* OR “adverse outcome pathway” OR Epigenetic* OR “DNA damage” OR “Biological effect” OR “Cellular effect” OR “Molecular event” OR “Key event” OR hepatic OR inflammatory OR immunity OR ROS OR “oxidative damage”)). The literature search was recently updated with relevant references from March 2020 to 30 July 2022, using the same search string, inclusion/exclusion criteria, etc., which are referred through each respective endpoint analyzed in Section 3. The resources searched included three databases of peer-reviewed literature: (i) Pubmed (all-fields and relevant MeSH terms); (ii) Scopus (only title-abstract-keywords: exclude conference papers, notes, editorials, and letters, etc.) and (iii) Web of Science (only core collection; exclude proceeding papers, meeting abstracts, news items, editorial material, and letters). The inclusion criteria were: (a) English spelling, (b) not a review, and (c) timeframe 2000–2020. Grey literature sources were not considered. The references were collected, managed, deduplicated, and screened using ZOTERO [17] and Microsoft Excel software.

### 2.1. First Stage Screening

In a first stage, relevance screening of the publications identified in the literature search was performed. The titles and abstracts were screened by two independent reviewers per reference with a third reviewer to solve any conflicts. All reviewers first screened the same 25 references to agree on how the screening criteria would be applied; additional subsets were independently screened. References were excluded once two reviewers agreed on exclusion. At this first stage, four questions were prepared in order to select papers: (1) Is it a review; (2) Does it concern TiO_2_-NPs; (3) Does it concern ingested NPs or have GIT targets (e.g., intestinal cells, GIT organs); (4) Does it include in vitro, in vivo, human volunteers or epidemiological data (e.g., Molecular, cellular events, bioavailability/bioaccumulation or adverse effects)?”. Questions for article screening at this stage are summarized in the format of a decision tree in Figure 1. References that fulfilled the inclusion criteria proceeded to full text screening, i.e., the second stage of screening.

### 2.2. Stage II Screening

In the second stage, before starting data extraction, leading team members curated the database for consistency. At this stage, it was decided to include only the mammalian models, excluding invertebrates and other in vivo models. Each complete article (i.e., full text) was screened by one reviewer, following standardized forms and guidelines for data extraction and for filling of the data extraction database. The complete list of the studies captured in the database is provided as Appendix A.

### 2.3. Setting up an AOP

At a final stage, a narrative synthesis of the review findings was generated qualitatively comparing the results from all studies. Putative AOPs were delineated based on this narrative, through interactive discussions in group meetings. Additionally, the AOP-helpFinder web server [18] was also used in order to confirm the novelty of the results, using the TiO_2_-NPs as a stressor and the effects defined on the first search string. With this tool, no results were retrieved, confirming the innovative putative AOP proposals. 

## 3. Results and Discussion

### 3.1. Overview of the Results

The literature search yielded a total of 1308 papers among the three databases used (Scopus, Web of Science, and Pubmed) with a total of 787 records screened at stage I. An overview of the papers obtained and the exclusion workflow is shown in Figure 2.

For Stage II, 282 papers were selected, with 97 being excluded for several reasons: considered non-mammalian models (64); not related with GIT (11); not nanosized (less than 100 nm) (9) and for other specific reasons (13), such as previously undetected reviews, not available in English version, etc. Therefore, 185 papers were eligible for data extraction and analysis (See Appendix A). Additionally, 34 papers were retrieved when the search was updated and were used for qualitative analysis. 

Concerning the source of the NPs, data extraction revealed that 33 papers (17.8%) did not provide any information on its provenience, although the majority referred to have acquired/obtained NPs from Sigma-Aldrich^TM^, St. Louis, MO, USA (n = 53) and 13 from the Joint Research Centre (JRC).

The majority of the papers (51.4%, 95/185) used sonication/ultrasonication as dispersion method used prior to biological assays application. Of notice, 42.2% (78/185) did not provide any information about dispersion methods.

Regarding the TiO_2_-NPs characteristics, several physicochemical parameters were provided. Although the crystalline phase is an important item on the characterization of NPs, 31.9% (59/185) of the studies did not provide information to this regard. The anatase (n = 81) and mixture of anatase/rutile (n = 39) were the most studied crystalline phases. Electron microscopy detection was used to analyze particles in 22 studies. Most papers used TiO_2_-NPs with sizes below 100 nm, but about 20% (36/185) of the analyzed papers were focused on TiO_2_ with more than 100 nm, and 12.4% (23/185) of the papers did not present further information about the NPs dimension, being used as provided by commercial sources. It should be emphasized that E171 also presents particles sized over 100 nm, and mixtures of different sizes are found in food. In spite these are not in line with the recognized NM definition and recommendations to describe properties, all of the papers were included for pursuing the analysis. Concerning the actual complexity of NP definition [5], it is possible that those papers could be considered out of the definition, but we decided not to exclude them. The information of specific surface area and charge of the TiO_2_-NPs was only provided in about 34% (63/185) of the analyzed papers. Forty-four studies presented a negative surface charge of the particles while only twenty-two presented positively charged particles (Table 1). 

From the total of 185 revised papers, the majority (67.6%, 125/185) reported in vivo studies. Among the in vivo murine models, the liver, blood, spleen, kidney, or GIT-related organs/tissues were the most frequently analyzed organs. The in vitro studies (37.3%, 69/185) were mainly focused in human cells, and the majority of them (70.4%, 38/54) developed with GIT-related cell lines. Fifteen in vitro studies were related to murine cell models. Only nine papers were based on humans (Table 2).

From the 69 in vitro studies analyzed, a significant percentage (65.2%, 45/69) were performed exclusively in vitro. However, in 24 studies, despite the assays being performed in vitro, the cells/tissues were collected from the animal models used in the experiments. Thus, these 24 studies considered oral gavage (10 studies, 41.7%), peroral (9 studies), and intraperitoneal (1 study) as methods for administering the substances to the animals. The considered studies presented a broad distribution of exposure duration. The lowest exposure was 15 min [19] and the highest corresponded to one year [20]. Concerning the dose range, quite heterogeneous doses were considered (50–100 mg/kg/day), reflecting the different approaches that were applied. A considerable number of studies (n = 123) did not report the methods used to characterize the NPs in the exposure medium used for the biological assessment. In the studies that did, Transmission Electron Microscopy (TEM) was the most used method (29 studies), followed by Dynamic Light Scattering (DLS, 22 studies) and Scanning Electron Microscopy (SEM, 15 studies). Regarding the characteristics of the NPs in the exposure medium, when reported, agglomeration of NPs was frequently described, reporting sizes higher than 100 nm.

Several biological endpoints of relevance for AOPs were identified among the 185 selected manuscripts (Figure 3). Predominant effects studied in the GIT and related organs (liver, spleen, kidneys) were oxidative stress (n = 73), cytotoxicity/apoptosis/cell death (n = 72), inflammation (n = 58), cellular and systemic uptake (n = 50), genotoxicity (n = 35), and carcinogenicity (n = 5). After data extraction, nine papers [21,22,23,24,25,26,27,28,29] were found to be not related with any of the 10 endpoints selected for analysis, although they were not excluded due to the fact of being possibly relevant for the discussion. 

The evidence of KEs and possible AOs of TiO_2_-NPs toward these endpoints is revised in the next sections, where a narrative synthesis compares qualitatively the findings from the selected studies. Additionally, results obtained from 2020–2022 were included in the discussion in each of the following sections but not included in the Appendix A presented. 

### 3.2. Molecular and Cellular Effects

#### 3.2.1. Cellular and Systemic Uptake 

In this study, a total of 50 papers described data related to the cellular uptake of ingested TiO_2_-NPs (see Appendix A). The ability of the TiO_2_-NPs to penetrate through the GIT was majorly studied in vitro by transwell systems (n = 33), and few in vivo (n = 19). Twenty-two studies observed the TiO_2_-NPs intracellularly by TEM/SEM or measured titanium content by ICP-MS/OES/AES methods (n = 19).

Translocation of particles through the intestinal barrier is a multistep process that involves diffusion through the mucus layer, contact with enterocytes and/or M-cells, and uptake via cellular entry or paracellular transport [30]. 

In several of the selected studies, the authors measured the transepithelial electrical resistance (TEER) and evaluated the expression of the epithelial membrane proteins (e.g., tight junctions, adherens junctions) to indirectly assess the epithelial barrier integrity after exposure to TiO_2_-NPs [31,32,33,34,35]. These authors postulated that a disruption of the epithelial barrier integrity would be the most likely mechanism through which TiO_2_-NPs could move to circulation from the intestinal lumen. Impaired GIT barrier was observed in vitro [31,36,37,38] and in vivo [39,40], in a dose-dependent manner. 

Regarding the expression of epithelial membrane proteins, as tight junctions (e.g., ZO-1) and adherens junctions (e.g., γ-catenin), different, and in some cases, contradictory results were reported. Some authors demonstrated that the expression of these proteins was not significantly affected by the TiO_2_-NPs exposure, in monocultures and cocultures of epithelial cells with other cell types [31,34]. However, in other studies, results showed that epithelial membrane protein expression was affected by the exposure to TiO_2_ NPs [41], which in the authors’ opinion, could justify the passage of NPs through the intestine by following through the paracellular route via disrupted tight junctions. The study from Talbot and colleagues [42] observed penetration of food-grade E171 TiO_2_ particles into the mucus and accumulation inside “patchy” regions within HT29-MTX cells, suggesting the absence of a mucus barrier impairment under “healthy gut” conditions. Additionally, in a study by Bettini et al., 2017 [3], TiO_2_-NPs were able to penetrate into and through the polarized epithelial cells without disrupting junctional complexes, as measured by γ-catenin levels. 

It has been argued that one of the most common mechanisms for uptake of NP into intestinal epithelial cells appears to be endocytosis [43]. Twenty studies were identified presenting results concerning the translocation of TiO_2_-NPs through the intestine. The translocation of TiO_2_-NPs was evaluated through different conditions in the studies considered in the present review. Notwithstanding, the selected studies encompassed all the analytical approaches used in the literature (i.e., in vitro, ex vivo, animal, and human studies). Most of the considered studies used in vitro approaches based on monocultures of polarized Caco-2 cells or co-cultures of Caco-2 with Raji B cells (which induce differentiation of a portion of Caco-2 cells to M-cells) in transwell systems [32,33,34,35,41,44,45,46,47,48]. Cabellos et al. (2017) [35] observed a higher translocation of TiO_2_-NPs through a Caco-2/M-cell model than in a Caco-2 monoculture model, suggesting that M-cells assume a relevant role in TiO_2_-NPs absorption. Janer et al. (2014) [46] also argue that the Caco-2 model would be improved by including M-cells in co-culture, considering the relevance of M-cells in the absorptive process, while in another study, undifferentiated Caco-2 cells internalized native NPs [44]. Interestingly, a strong interaction between TiO_2_-NPs and mucin has been shown, indicating that mucin absorb to the surfaces of the TiO_2_-NPs and reduce their tendency to aggregate [49]. In buccal epithelial cells, TiO_2_-NPs were able to bind to the cellular membrane and pass into the cells in a dose dependent manner [50]. Results pointed out that the translocation of these NPs is scarce [31,33,41,47,51,52], frequently not exceeding 1% of the exposure dose. MacNicoll and colleagues [47] investigated the uptake and biodistribution of nano- and larger-sized TiO_2_, using the Caco-2/M-cell in vitro model of human gut epithelium, and in vivo in rats, showing that oral administration of 5 mg/kg body weight of TiO_2_ nano- or larger particles did not lead to any significant translocation of TiO_2_ either to blood, urine or to various organs in rats over a 96 h post-administration period. It has also been shown a very low oral bioavailability and slow tissue elimination of TiO_2_-NPs [53]. Only Koeneman and colleagues [31] reported a higher percentage of NPs moving from apical to basolateral chambers (14.4% of the exposure dose), which was also evidenced by the in vitro studies by Veronesi et al. (2012) [48] using the Caco-2/M-cell co-culture model. Jo et al. (2016) [45] investigated the in vivo oral absorption of food-grade TiO_2_-NPs (f-TiO_2_-NPs) compared to general grade (g-TiO_2_-NPs), and the intestinal transport pathway, using an in vitro approach, showed that most of the NPs were eliminated through the feces. Other authors also reported that dietary TiO_2_-NPs are likely to be excreted in the feces [47,54,55]. 

The absorption of TiO_2_-NPs from the gastrointestinal tract was also indirectly assessed by measuring Titanium contents into secondary organs. Titanium was detected in blood, brain, lungs, heart, kidney, liver, spleen, pancreas, testicles, and small and large intestine [45,56,57,58,59,60,61,62,63,64]. TiO_2_ occurred in some of these organs in a NP size-dependent manner [63]. In addition to these organs, a significant increase in the Titanium contents was detected in the maternal serum, placenta, and fetus, suggesting that TiO_2_ can cross the intestine, infiltrate the maternal blood and move across the placental barrier reaching the fetus [65]. Furthermore, it was shown on mice that depending on dose, TiO_2_-NPs ingestion can cause the destruction of dopaminergic neurons and consequently increase the risk of Parkinson’s disease [66]. 

More recently, it was reported that dietary nanoparticles compromise epithelial integrity on human intestinal epithelial cells (Caco-2 and HIEC-6) [67]. Another work showed that long-term intake of food additive TiO_2_ in ICR mice altered the intestinal epithelial structure, however, without influencing intestinal barrier function [68]. Furthermore, an in vivo and ex vivo study in mice showed that jejunal villus absorption and paracellular tight junction permeability are major routes for early intestinal uptake of food-grade TiO_2_ particles [69]. In addition, the oral administration of TiO_2_-NPs seems able to induce intestinal inflammation and destroy the integrity of intestinal barrier [70]. Based on these studies, it seems clear that TiO_2_-NPs ingestion significantly change the intestine physical barrier in a dose-dependent manner [71], and these particles can enter in the systemic compartment and accumulate in several organ.

#### 3.2.2. Oxidative Stress

This review identified 73 studies concerning ROS generation or effects on oxidants and antioxidants markers, following exposure to TiO_2_ (Appendix A). Those markers included reduced and oxidized forms of glutathione (GSH/GSSG), malondialdehyde (MDA, a marker of lipid peroxidation), enzymatic activity of glutathione peroxidase (GPx), superoxide dismutase (SOD), and catalase (CAT). Among those studies, 35 were in vitro and 48 in vivo. Four papers included both in vitro and in vivo data [72,73,74,75]. Only one study was performed in humans [36], focusing on the determination of the food-derived TiO_2_-NPs effects on compromised epithelial barrier, showing increased levels of titanium in blood of patients with acute colitis. 

In vivo, different rodent models were used, and the considered markers of oxidative stress comprised lipid peroxidation products [malondialdehyde, MDA], reduced GSH content, oxidized GSSG content, GSH/GSSG ratio, SOD, GPx, CAT, sulfhydryl groups (SH), total oxidant status (TOS), total antioxidant status (TAC), nitric oxide, superoxide anion, sulfhydryl groups (SH), and GSTT. In vivo, 35 studies reported increased oxidative stress after oral exposure to different doses of TiO_2,_ being that a dose-dependent increase was reported by three studies [72,76,77]. Oxidative stress biomarkers were analyzed in different organs, but the majority were focused in the liver (27 studies). 

It was demonstrated in vivo that oxidative stress could be regarded as a key player in TiO_2_-NPs induced liver injury [78,79]. It was observed that mouse serum levels of the liver enzymes ALT, AST, and ALP increases significantly, which prompts cellular leakage and loss of functional integrity of liver cell membranes. Additionally, oral administration of TiO_2_-NPs causes a significant rise in the hepatic levels of ROS along with a significant reduction in GSH levels. Thus, TiO_2_-NPs have a tendency to generate free hydroxyl radicals leading to genotoxicity induced by oxidative stress and ultimately apoptosis [76,78]. In vitro, the preferred method used to ascertain oxidative activity was the quantitative cell-based 2′-7′dichlorofluorescin diacetate assay (DCFH-DA) or derivatives, performed in 22 papers (Appendix A). 

In vitro studies addressing ROS generation and/or oxidative stress were mostly performed in human cell models (29/35), mainly in intestinal cell models such as polarized and nonpolarized Caco-2 cell monolayers. Overall, most in vitro studies in human cell models (21 out of 28) and in murine cells (3 out of 4) revealed increased oxidative stress generation upon NPs exposure. In Caco-2 cells, eight studies showed a positive effect [36,44,72,75,80,81,82,83], while six studies reported negative results [73,84,85,86,87,88]. Additionally, a recent in vitro study [89] also showed the absence of ROS induction in Caco-2 and HT29-MTX-E12 cells exposed to the TiO_2_-NPs, suggesting that at physiologically relevant concentrations for the human intestine [39,90], these effects are of no concern. Another study using Caco-2/HT29-MTX co-cultures, also reported no induction of ROS generation, after 4 h exposure to a single dose of 0.14 μg/mL of anatase TiO_2_ (30 nm), although these authors observed damage to epithelial microvilli and decreased glucose absorption [39]. It was described that the physiological constituents present within the GIT can alter the physiological parameters of TiO_2_-NPs such as pH, ionic strength, as well as protein content and composition [44], and, therefore, the interaction with cells. A publication from Cao and colleagues [91] showed ROS generation after exposure to digested food models containing E171 (110 nm) in the Caco-2/HT29-MTX co-culture model and found increased ROS generation. 

In a recent publication with rats that orally ingested TiO_2_-NPs, it induced tissue-specific oxidative stress in liver and imbalance of elements [92]. Depleted lipid peroxidation levels and protein carbonyl content, in mitochondria, have also been induced by TiO_2_-NPs in hepatic cells [25,93]. In fact, several publications mention morphological changes in mitochondria [25], swelling [40,56,94,95], damaged membranes [96,97,98], and causing decreased mitochondrial activity [99] in several tissues. Moreover, the presence of TiO_2_-NPs in lysosomes in liver [100] and renal tissue [25] has been previously recognized. In conclusion, considering that few studies reported no effect of exposure to TiO_2_ in the analyzed biomarkers [59,75,101,102,103], this suggests that oxidative stress is a key cellular event driving TiO_2_-NPs biological effects.

#### 3.2.3. Cell Death and Proliferation

We have identified 72 studies concerning the cytotoxicity/apoptosis/cell death endpoints (see Appendix A). Out of these 72, 38 were in vitro approaches and 21 in vivo studies. Thirteen included both in vitro and in vivo data. No studies in humans were reported.

For in vitro cellular proliferation and viability evaluation, different methodologies were used such as: (a) cell metabolic activity-based tests using tetrazolium salts (MTT, the most common: 11; WTS: 7; MTS: 2) or rezasurin [104]; (b) cell membrane integrity tests (LDH: 13, dye uptake, neutral red: 2, dye exclusion, tryptan blue: 4 and propidium iodide: 1). Colony forming efficiency [80], flow cytometric techniques [105] and Vialight Plus bioluminescence assay Kit [37] were used in some studies. For apoptosis analysis, techniques like flow cytometry [104,106], using dyes such as Annexin V staining [97,99,106,107,108], TUNEL assays [60,79,109,110,111,112], and caspase activity detection kits [76,78,113,114] were applied. The majority of the studies used monocultures of human Caco-2 cells (23/72). In four studies [34,41,81,91], a coculture of Caco-2 with HT29-MTX was used. Another study used a 3D intestinal model, consisting of Caco-2 cells and two human immune cell lines [115]. 

Most papers (47/72) revealed increased cytotoxicity upon NPs exposure, whereas 18 did not show any effect. A dose-dependent effect was indicated in 16/72 [72,82,104,109,113,116,117,118,119,120,121,122,123,124,125,126]. Furthermore, Gandamalla et al. [82] showed that not only TiO_2_-NPs dose, but also its size, could influence its potential toxicity. The study concluded that smaller sized TiO_2_-NPs (18 nm) induced greater toxicity at lower concentrations than the bigger sized NPs (30 and 87 nm), owing to their varying physicochemical properties. A concomitant increase in ROS levels was also detected. 

Moreover, in vitro exposure of human Caco-2 cells to pure anatase TiO_2_-NPs of primary particle size of 100 nm resulted in reduced cell viability compared with rutile nanoparticles of 50 nm, suggesting that TiO_2_-NPs toxicity in human intestinal cells depends on the particle size and crystalline structure [83]. Natarajan et al. [119] found not only a concentration (0–1000 ppm) but crystalline type dependent loss in primary hepatocytes viability after exposure to 3 different TiO_2_-NPs, P25 (21 nm; LC50 = 74.13 ± 9.72 ppm), pure rutile (50 nm; LC50 = 58.35 ± 4.76 ppm) and pure anatase (50 nm; LC50 = 106.81 ± 11.24 ppm). Chakrabarti, et al. [104] showed that cell viability decreased with increasing doses of TiO_2_-NPs in both in vitro and in vivo experiments. Furthermore, higher doses resulted in more severe oxidative damage and eventually imposed cell cycle arrest and apoptosis of the damaged cells. The functionalization of TiO_2_-NPs is another important factor that has influence in the cytotoxicity of this material. TiO_2_–core nanorods (NRs) and TiO_2_–NH_2_ NRs reduced cell viability more than those of TiO_2_–COOH NRs and TiO_2_–PEG NRs, after 72 h exposure of rat bone marrow stem cells [120]. 

Increased cytotoxicity may be associated with a similar trend in other endpoints as apoptosis [72,104], ROS [72,82], genotoxicity [104] or inflammation [121]. Nevertheless, this is not a general rule. For example, an increase in cytotoxicity may not correspond to an increase in genotoxicity, as was observed in murine cells after exposure to TiO_2_ anatase of 20 nm (20 μg/mL) [122]. Also, no cytotoxicity effect was observed in human gingival fibroblasts after exposure to TiO_2_-NPs [123]. Furthermore, NM-105 increased ROS production in human buccal epithelial cells without affecting cell viability/integrity [124]. Bettencourt and colleagues [125] showed that digested NM-105 presented a more pronounced toxicity in HT29-MTX-E12 intestinal cells, as compared to undigested NPs. Additionally, TiO_2_-NPs had no obvious effect on cytotoxicity but induced a strong immune response in murine macrophages (RAW 264.7 cell line) [126]. 

Finally, there are some ambiguous cytotoxicity results. Some of the collected reports describe a lack of toxicity after TiO_2_ exposure in spite of its uptake and translocation through the cells without disrupting junctional complexes or epithelial integrity [31,44,110,118]. A very recent study showed that, although no clear effects on cytotoxicity were observed following repeated exposure of differentiated Caco-2 and HepaRG cells to TiO_2_-NPs, subtle effects on membrane composition could induce potential adverse effects in the long-term [127]. On the other hand, there are results supporting that the toxicity attributed to TiO_2_ is related with its cellular accumulation, which may potentially lead to the dysregulation of cell function and cell death [37,47,51,120]. In studies with normal colon cells (CCD-18Co) in vivo and in human colon organoids, it was shown that TiO_2_-NPs induce cytotoxicity in two-dimensional CCD-18Co cells and three-dimensional CCD-18Co spheroids and human colon organoid [128].

#### 3.2.4. Cell Signaling

We have identified 12 studies concerning changes in cellular signaling pathways upon exposure to TiO_2_-NPs. Out of the 12, only two are in vitro studies, one of which is in primary human cells, the remaining studies are in mice. Four of these 10 in vivo studies are complemented with information from in vitro data.

In general, the tests with repercussions on cell signaling, both intracellular signaling and cell to cell communication, are done with a maximum of 21 days of oral exposure to the NPs, which in most cases is pure anatase. Signaling is deregulated at the level of the immune system, via inflammasome activation [8,129], olfactory signaling pathway [8], cell proliferation [130], apoptosis [131], increase in SDF-1 pathway (involved in acute liver injury and subsequent tissue regeneration [58], and increased apoptosis through upregulation of Bax and downregulation of Bcl-2 proteins [132]. 

The use of anatase in in vitro studies in human primary periodontal ligament cells (PDL) showed a change in intracellular signaling pathways, such as activation of ERK1/2 and AKT. Concomitantly, ROS overproduced in response to TiO_2_-NPs induce COX-2 expression through activation of NF-κB signaling, which may contribute to the inflammatory effect of PDL cells [121]. Furthermore, anatase could inhibit the growth of lung cells and allow a considerable proportion of the cells with TiO_2_-NPs in the cytoplasm to remain in the G1/G0 phase. On the other hand, in vivo studies in mice subjected to oral exposure of anatase, pure or mixed with rutile, show a downregulation of genes involved in the innate and adaptive immune system [130]. 

Although several of the collected studies report that TiO_2_-NPs can enter cells, pass through biological membranes like blood–brain and placental barriers, accumulate in tissues and organs, and trigger multiple cellular responses, the available data on the changes in cellular signaling that enable these effects is still limited. 

#### 3.2.5. Inflammation

From 58 collected studies concerning inflammatory endpoints, 35 were conducted in rodent models by oral TiO_2_-NPs administration (exposed from 5 days to 26 weeks), 21 in cell lines (mainly Caco-2 or macrophages, but also some hepatic, bronchial, or gingival cells, exposed for 3–72 h), and 12 included both (in vitro and in vivo) models. The main inflammation-related results from these studies, were organized in six topics: effects on immune cells (12 studies), changes in cytokine levels (32), colon inflammation (14), effects on other cells/organs (43), oxidative stress (30) and gut microbiota changes (5). 

Concerning the effect on immune cells, exposure of macrophages, THP-1 monocytes or dendritic cells to TiO_2_-NPs (10–50 nm) was generally found to promote the expression of pro-inflammatory cytokines such as interleukin (IL)- 1-β, IL-6, or IL-8, an effect frequently accompanied by increased ROS-production and inflammasome activation [36,50,75,78,83,106,122,133,134,135,136,137]. Overall, these studies indicated an imbalance of the immune response, including macrophage activation, reduced Treg cell-mediated inflammatory control, lower chemotactic or bactericidal activities and higher susceptibility to infection by gastroenteritis norovirus (MNV-1) [3,122,134,138]. Only one study reported no differences in the percentage of dendritic, CD4+ T or Treg cells within Peyer’s patches, or in cytokine production [139]. 

Several of the collected studies report that, upon rodent oral exposure to TiO_2_-NPs, increased concentrations of pro-inflammatory cytokines, including IL-1α, IL-4, IL-6 and TNF-α, were detected in serum [58,60,78,94,114,139,140,141,142,143,144,145], or in tissues such as liver, spleen, kidney, intestine, testis or brain [7,58,59,94,114,146,147,148,149,150,151]. Exposure of intestinal [34,37,80,83,109,152,153] or gum [50,121,143] cell models also reported increased secretion of pro-inflammatory cytokines by epithelial cells. In regard to colon inflammation, some studies reported that TiO_2_ is not inert, but rather impairs gut homeostasis, which may in turn either prime the host for disease development, including colitis-associated cancer, or worsen a pre-existing bowel disease. Administration of TiO_2_-NPs induced pro-inflammatory changes and reduced intestinal barrier function in the gut [7,36,72,90,133,138,154,155,156], especially in mouse models for acute colitis. Some adverse effects in other organs have also been reported. One study using low TiO_2_ concentrations (~2 mg/kg/bw) found no significant disturbance of the function or structure of the intestine, liver, spleen, lungs, or testis [139]. After rodent oral exposure to TiO_2_-NPs, inflammatory reactions with tissue morphological or functional changes were observed in several organs, including spleen [36,112], liver [56,58,60,72,77,78,94,140,142,143,148,157,158,159] (with increased serum glucose and insulin resistance, or fibrosis and increased serum levels of liver enzymes), lung [160,161], kidney [145,151] (altered urea, creatinine and uric acid levels), heart (with changes in heart rate and systolic blood pressure) [144], testis [114,149], brain [150], stomach [146,162], and transfer of TiO_2_-NPs to offspring via breast milk [111]. 

In many studies, increased ROS were reported, being regarded as key players in TiO_2_-NPs induced tissue injury or inflammation in the liver [60,77,78,94,110,140,142,143,147,148,158], intestine [8,58,90,154], gum [121] and colon cell lines [36,80,109,163]. Accordingly, added antioxidants revealed protective effects [60,78,121,143,145,147,149,158]. Recent results showed that oral administration of 200 mg/kg TiO_2_-NPs induced only modest changes in liver function parameters, but could induce intestinal inflammation [70]. 

In addition, metabolic disorders of gut microbiota with subsequent lipopolysaccharides (LPS) production in response to oral exposure to TiO_2_-NPs led to oxidative stress and an inflammatory response in the intestine [94,154]. However, studies using only low TiO_2_ concentrations (~2 mg/kg/bw) found no significant increase in the intestinal permeability or disturbance of gut microbiota [58,139,164,165]. Other studies, reported additional changes in the gut microbiome and its metabolism [94,138,154,166,167]. In a murine model, TiO_2_ led to changes in gut microbiota, especially mucus-associated bacteria [168]. Long-term oral exposure to dietary NPs at doses relevant for estimated human intakes disrupted the gut microbiota composition and function [169]. Other recent studies showed that TiO_2_-NPs ingestion altered the GI microbiota and host defenses promoting metabolic disruption and subsequently weight gain in mice [170], and that it leads to adverse disturbance of gut microecology and locomotor activity in adult mice [171]. However, one study did not find major effects of dietary exposure to the TiO_2_-NPs on the murine gut microbiome [172]. 

In conclusion, the ingestion of TiO_2_-NPs triggered signs of inflammation and production of proinflammatory cytokines, frequently accompanied by increased ROS-production. The above-mentioned studies suggest the induction of imbalanced immune responses, particularly in the presence of pre-existing inflammatory conditions.

#### 3.2.6. Genotoxicity

We have identified 35 studies concerning the genotoxicity of TiO_2_-NPs in the GIT. Most of the studies (25/35) revealed increased genotoxicity upon NPs exposure, although some (10/35, 6 of which including in vivo studies) did not find evidence of such an effect (Appendix A). 

In the 25 studies reporting genotoxicity of TiO_2_-NPs in GIT, anatase, rutile or their mixture was used, and 14 studies included in vivo assays. For example, the work from Chen et al. [173] evaluated the genotoxicity of anatase TiO_2_-NPs (75 ± 15 nm) in vivo, revealing that the doses of 50 and 200 mg/kg body weight, every day for 30 days, induced DNA double strand breaks in bone marrow cells, but did not induce damage to chromosomes or mitotic apparatus observable by the micronucleus assay [173]. DNA damage was observed at the concentration of 100 µg/mL after 24 h treatment using the comet assay, while induction of gene mutations was observed at the concentration of 20 and 100 µg/mL after 2 h treatment using hypoxanthine-guanine phosphoribosyl transferase (HPRT) gene mutation assay, overall revealing that TiO_2_-NPs can induce genotoxic effects both in vivo and in vitro tests [173]. 

There were 18/35 reports showing DNA damage induction using the comet and/or fpg-modified comet assay, 11 of which were in vivo studies. This seems to be an event frequently correlated with ingested TiO_2_-NPs, although as genotoxicity biomarker it is not an endpoint clearly linked to health outcomes, since the detected DNA single- or double-strand breaks are primary lesions that may be repaired by the cell repair machinery, or may lead cells to programmed cell death [174]. Within this review, 9 authors reported the use of this biomarker, where 6 report its increase upon TiO_2_-NPs exposure in intestinal cells. That was the case of the anatase/rutile E171, leading to MN increase in human colon (HCT116) cells, as well as DNA damage [85]. In rats, intragastric administration of anatase TiO_2_-NPs for 60 days at 100 and 200 mg/kg body weight led to micronucleus induction in rat bone marrow, and DNA damage by the comet assay was also observed [175]. Also, mouse erythrocyte bone marrow micronucleus test showed a significant increase at the highest dose (100 mg/kg) of anatase TiO_2_-NPs, after 14 days of oral exposure, while DNA damage was increased at all concentrations [76]. A high incidence of micronucleated red blood cells was reported upon oral exposure of rats to an unspecified form of TiO_2_ [57]. TiO_2_-NPs induced both DNA damage and micronuclei in bone marrow cells of mice exposed orally to anatase 1000 mg/kg daily [176]. Furthermore, the in vivo micronucleus and chromosomal aberration assays, both performed according the OECD guidelines, showed and increase at the dose of 500 mg/kg of TiO_2_-NPs administered for 90 days in mice, but no increase at lower concentrations, while increased comet tail length was also observed with the higher doses of TiO_2_-NPs [104]. Increased chromosomal aberrations were observed in mice after five days of oral administration of TiO_2_ in doses of 250 and 500 mg/kg body weight [77] and after gavage-mediated exposure of mice to rutile TiO_2_ at sub-acute concentrations (0.2, 0.4 and 0.8 mg/kg body weight) over a period of 28 days [177]. In spite of the 3 reports of no effect of ingested TiO_2_ in chromosomal damage [86,157,173], it appears as a probable event upon oral exposure to TiO_2_ that deserves further research. In a recent study [89], the in vitro results in Caco-2 and HT29-MTX-E12 cell lines evidenced a DNA-damaging effect dependent on the NP, more relevant for the rutile/anatase NM-105, possibly due to its lower hydrodynamic size. Moreover, micronucleus assay results suggest an effect on chromosomal integrity, in the intestinal HT29-MTX-E12 cell line exposed to TiO_2_-NPs (NM-102, NM-103, and NM-105), especially after an in vitro digestion procedure. Of particular concern may be chronic exposure, even at high doses of 500, 1000 or 2000 mg/kg body weight of several TiO_2_ [157]. 

An effect on the frequency of gene mutations was observed in vivo at the concentrations of 20 and 100 µg/mL, after 2 h treatment in the OECD-compliant HPRT gene mutation assay [173], while other authors reported negative results in the same assay [96]. Conversely, mice orally exposed to 5, 50 or 500 mg/kg body weight TiO_2_-NPs for five consecutive days presented high mutation frequencies in p53 exons 5–8 in a dose- and time-dependent manner [162]. Therefore, further assays are necessary to address the induction of gene mutations upon exposure to ingested TiO_2_-NPs. 

The only in vitro study retrieved that attempted to address TiO_2_-NPs carcinogenicity characterized the effect of commercially available NPs (P-25, 21 nm, 80/20 anatase/rutile) and nanopowder 637254 (titanium (IV) oxide anatase, <25 nm) in human gastric epithelial cells [178]. The authors found that TiO_2_-NPs induced oxidative stress and genotoxicity that could mediate the observed uncontrolled cell proliferation and apoptosis evasion, which are hallmarks of tumor cells [178]. 

An ex vivo exposure of peripheral blood lymphocytes from gastrointestinal disease patients revealed a concentration dependent induction of DNA damage, by the comet assay, and the frequency of micronuclei (MN) in binucleated cells was increased in a concentration-dependent manner [179], suggesting that the relation of genotoxic effects of TiO_2_ with individual susceptibility or pre-existing diseases is also a matter of concern. 

Overall, we conclude that the most frequent molecular event regarding a genotoxic effect of ingested TiO_2_ was DNA damage, usually detected by the comet assay, followed by chromosomal damage.

#### 3.2.7. Carcinogenicity

Amongst the studies retrieved, seven were identified as related to the carcinogenic potential of ingested TiO_2_ [3,7,8,19,129,160,178]. All but one of the studies used rodent models orally exposed to TiO_2_-NPs for different periods of time and reported results from several endpoints at the tissue, cellular and molecular levels, attempting to understand the mechanisms behind these NPs’ carcinogenic effects, as well. Ammendolia et al. [19] tested the in vivo and in vitro genotoxic and carcinogenic effects of TiO_2_-NPs (anatase, primary size < 25 nm, BET surface area 45–55 m^2^/g, purity 99%), whereas the other carcinogenicity studies used experimental animals only and tested the food grade E171 [3,7,8,129]. 

Four in vivo studies tested food grade TiO_2_ in rodents, producing positive results as to its carcinogenicity and advancing the knowledge of the underlying mechanisms of action [3,7,8,129]. Urrutia-Ortega and colleagues [7] showed a significant enhanced tumor formation in the distal colon of a chemically induced colitis-associated colorectal cancer (CRC) mouse model, after intragastric administration of E171 (5 mg/kg body weight) for 10 weeks. CRC progression and inflammation markers indicated that E171 exacerbates tumor progression and inflammation. In another study [8] from the same group, analysis of initial transcriptome changes in colon tissue before the neoplastic alterations appeared, showed that intragastric exposure to the same concentration of E171 for 2, 7, 14, and 21 days led to the upregulation of genes involved in activation of inflammation, reduction of immune capacity, and both up- and downregulated genes involved in development of cancer, for instance of colon cancer [8]. The results were in line with previous studies in which oxidative stress and DNA damage was observed in vitro in colon epithelial cells after E171 exposure [85,178]. Bettini and colleagues [3] observed, after ingestion of E171 by rats, a potent Th1/Th17 immune response via an increased production of IFN-γ in Peyer’s Patches and IFN-γ and IL-17 in the spleen after 7 days of exposure. In addition, using a CRC mouse model the same authors showed that E171 exposure for 100 days induced a release of inflammatory molecules, preneoplastic lesions as well as the growth of aberrant crypt foci. The observed effects are in agreement with the tumor formation in this CRC model [129]. Interestingly, E171 affected genes involved in biotransformation of xenobiotics which can form reactive intermediates increasing their toxicological effects. 

Ammendolia and colleagues [19] investigated potential modulatory effects of low doses of the above mentioned TiO_2_-NPs (2 mg/kg bw per day or 1 mg/kg bw per day or vehicle) on intestinal cells of adult Sprague-Dawley rats treated by gavage for 5 consecutive days. The authors suggested that TiO_2_-NPs deposition in intestinal cells, as detected by ICP-MS determination of titanium, might have induced hyperplasia, likely related to increased villi size observed. Mechanistic studies were performed with HT29 cells exposed to the same TiO_2_-NPs (1 and 5 mg/cm^2^) revealing neither cytotoxicity nor ROS production upon TiO_2_-NPs exposure, although uptake of NPs was detected by electron microscopy. If sustained, this effect could lead to an increased risk of tumor development or to progression of existing tumoral lesions. 

Very recently, using an Apc-gene-knockout model, which spontaneously develops colorectal tumors, E171 exposure induced an increase, statistically nonsignificant, in the number of colorectal tumors in these transgenic mice, as well as a statistically nonsignificant increase in the average number of mice with tumors, while modulation of events related to inflammation, activation of immune responses, cell cycle, and cancer signaling were shown by whole-genome mRNA analysis [180]. Conversely, food-grade titanium dioxide (E171) induced adenomas in colon and goblet cells hyperplasia in a regular diet model and microvesicular steatosis in a high fat diet model in mice [181]. 

Altogether, the reviewed studies suggest that chronic oral exposure to TiO_2_-NPs and, may aggravate and, if not initiate, at least promote the development and progression of preneoplastic lesions in the colon.

#### 3.2.8. Biochemical and Other Physiological Parameters

The present work identified 13 studies concerning the alterations in biochemical and other physiological parameters after the oral exposure to TiO_2_-NPs (see Appendix A). Several effects were considered in these studies: liver damage, cardiac damage, nephrotoxicity, hematological effects (blood cells count and coagulation parameters), lipid metabolism, glucose metabolism and reproductive toxicity. Some of these effects were considered simultaneously in the studies. Few studies demonstrated no systemic toxicological effects associated with the agglomerated/aggregated TiO_2_ P25 during the repeated-dose 28-day, 90-day, and recovery studies in rats, and the substance was not detected in the target organs [182]. The subchronic toxic responses of E171 were studied using rats and AGS cells, a human stomach epithelial cell line and a NOAEL for 90 days repeated oral administration was set between 100 and 1000 mg/kg for both male and female rats [183]. 

Regarding liver damage, the enzymes’ serum levels (alkaline phosphatase-ALP, alanine aminotransferase-ALT, aspartate aminotransferase-AST, high density lipoprotein-ALT/AST ratio), albumin-ALB, lactate dehydrogenase-LDH and bilirubin-BIL were the main biomarkers used to assess the effects of exposure to TiO_2_-NPs in the liver. Significant effects in serum levels of ALT, AST, ALT/AST ratio, BIL or LDH were reported [40,56,72,78,143,184,185] as well as in loss of urea and albumin synthesis function of hepatocytes [119]. Two different studies reported not only the effects of TiO_2_-NPs in liver enzymes but also the protective effect of idebenone, carnosine, vitamin E and vitamin A [78,143]. In rats orally exposed, liver was the most sensitive tissue to TiO_2_-NPs-induced oxidative stress, showing decreased reduced glutathione (GSH), increased oxidized glutathione (GSSG) and decreased ratio of GSH/GSSG as well as accumulation of lipid peroxidation (malondialdehyde, MDA) in liver tissues, in a significant time-dependent relationship [92,186]. 

Regarding cardiac damage, the parameters creatine kinase-CK, LDH and HBDH were evaluated after oral exposure to TiO_2_-NPs. The significant modifications of the exposed groups when compared with control groups were considered as indicative of cardiac damage [40,184,185]. 

The toxic effects on kidneys were also evaluated and the significant differences for creatinine and urea serum levels between exposure and control groups were considered as indicative of kidney damage and attributed to the small size and difficult clearance of TiO_2_-NPs [72,184,185]. 

In respect to effects on hematological parameters, such as blood cells count and coagulation tests, the exposure to TiO_2_-NPs significantly increased the white blood cells count and red blood cells count, being this attributed to the activation of the immune function and inflammatory response, and to an adaptive body response to the toxic effects of TiO_2_-NPs, respectively) [40,185]. No significant alteration of coagulation parameters was reported [185]. In a recent study, E171 decreased hematocrit and hemoglobin in male but not in female mice while leukocyte and erythrocyte count remained unaltered [187]. 

Significant alterations of lipid and glucose metabolism were however reported after exposure to TiO_2_-NPs [188,189]. The prostatic and testicular toxicity associated with oral exposure to TiO_2_-NPs was also assessed and a significant effect on serum levels of prostate specific antigen-PSA, prostatic acid phosphatase-PAP, free testosterone-TST, estradiol-E2, luteinizing hormone-LH, follicle stimulating hormone-FSH was reported. The treatment with morin was determined as presenting a potential beneficial role, probably being mediated by redox regulatory, anti-inflammatory, and antiapoptotic mechanisms [114]. A study from 2019 [190], focused on the effects of E171 consumption on mice, showed increased germ cell sloughing and inflammatory cells, together with the disruption of the blood–testis barrier. Abnormal developmental events in male rat seminal vesicles have also been shown [191], and leading to spermatogenesis disturbances [192]. On the other hand, in a study from Hong et al. (2016), the exposure of female mice resulted in premature ovarian failure, which was triggered by alterations in hormones and autoimmunity markers [193]. 

In a recent study, oral administration of TiO_2_-NPs to rat pups impacted basic cardiac and neurobehavioral performance, neurotransmitters and related metabolites concentrations in brain tissue, and the biochemical profiles of plasma [194]. Furthermore, TiO_2_-NPs-induced neurotoxicity regarding AChE, serotonin, MDA, GSH, GPx, GST, IL-6, caspases-8, -9, and -3 in rats upon oral exposure [195]. TiO_2_-NPs exposure induced alterations on mice ovary resulting in a decrease in the rate of embryo development and fertility [196]. Lactating Wistar rats orally exposed to TiO_2_-NPs demonstrated significant impaired memory and learning in the offspring [130,197]. However, in vivo publications present heterogenous results on the development effects due to the ingestion of TiO_2_-NPs during pregnancy, such as the study from Lee and colleagues [198], which indicates increased Titanium levels in the maternal liver, maternal brain and placenta, but these levels did not induce marked toxicities in maternal animals or affect embryo–fetal development. Furthermore, Warheit and colleagues [199] found no evidence of maternal or developmental toxicity at any TiO_2_-NPs tested dose.

#### 3.2.9. Transcriptomics, Epigenomics and Proteomics

Only two transcriptomics [8,74], one proteomics [91], and one epigenomics [200] study were found on the present search. In one of the transcriptomics studies, significant changes in the expression of 139 genes were identified in mice exposed to 50 mg/kg TiO_2_-NPs for 26 weeks that suggested induction of endoplasmic reticulum (ER) stress, which in turn promoted the generation of ROS by activating the monooxygenase system. In turn, ROS played a key role in the induction of insulin resistance, triggering hyperglycemia in mice. Furthermore, TiO_2_-NPs stimulated the expression of CYP enzymes, such as Cyp4a14 and Cyp2b9, which are crucial in xenobiotic metabolism [74]. The other study focused on the influence of E171 exposure in the induction of inflammatory, immunological and specific cancer-related pathways in colon tissue of mice. E171 induced the activation of the immune response, oxidative stress, inflammation, GPCR/olfactory receptors, cell cycle, DNA repair, cancer related genes, metabolism and also serotonin receptors genes, which may facilitate the development of colorectal cancer [8]. 

Cao et al. (2020) [91] studied changes to the proteomic profile in a small intestinal epithelium tri-culture cellular model (Caco-2/HT29-MTX/Raji B cells) after exposure to digested food models containing E171 (110 nm). Liquid chromatography coupled with tandem mass spectrometry (LC-MS/MS) was used to analyze the cellular proteome and resulted in the identification of 4944 proteins with similar overall patterns of abundance between the food model control and TiO_2_-treated samples, suggesting that TiO_2_ induced a minimum impact on the cellular proteome [91]. 

As to epigenomics, TiO_2_-NPs caused global DNA hypomethylation in liver tissue samples of male rats that received TiO_2_-NPs by oral administration (100 mg/kg) for 6 weeks [200]. A significant decrease in the mRNA levels of SOD, CAT, GSHPx, MT and HSP70, CYP1A1, p53, GST, and TF genes, and an increase in CYP1A was also observed in livers of mice exposed to 10 or 50 mg/kg nano-TiO_2_ for 60 days, supporting that the TiO_2_-NPs liver toxicity is caused by damaged mitochondria, ROS generation, and changes in the expression of protective genes [98]. Among a variety of metabolic and transporter genes, up-regulation of the uptake transporter gene Oapt1, the basolateral efflux transporter gene Mrp3, and Cyp2b, which is involved in the metabolism of endogenous and exogenous compounds, were identified in mice exposed to TiO_2_-NPs (21 nm) for 14 days, suggesting disruption of bilirubin homeostasis [56]. Additionally, an increase in gene expression of BAX, caspase-3, and P53, and decrease in Bcl-2, SOD, GPx, CAT, and GSH, with a marked increase of gene expressions of NLRP3, caspase-1, IL-1β, TNF-α, and iNOS were reported in the intestine and liver of rats after 30 days exposure to 10, 50 and 100 mg/kg TiO_2_-NPs, strongly accompanied by intestinal oxidative stress, inflammation, apoptosis, and histopathological changes [72]. By contrast, there were no differences in the mRNA levels of the oxidative stress marker genes heme oxygenase-1 (HO-1) and γ-glutamylcysteine synthetase (γ-GCS) on Caco-2 cells exposed to five different TiO_2_-NPs [87]. No significant changes were also identified in the expression of 16 genes involved in ROS regulation, DNA repair via base-excision repair, and ER stress on a Caco-2:HT29-MTX co-culture exposed to E171 anatase (12 nm) and anatase/rutile (24 nm) TiO_2_-NPs [201]. These authors further reported a moderate change in gene expression of markers of intestinal epithelial differentiation, i.e., CLDN1, OCLN, TJP1, CTNNB1 (involved in adherens and tight junction), and SI and ALPI (involved in microvilli differentiation), without altered enterocytic differentiation; increased mucins, although the mRNA levels were either unchanged or moderately down-regulated [153]. The results evidenced a moderate dysregulation of markers of the intestinal barrier function, which enhanced a protective response of the epithelium. Shrestha et al. (2016) [120] studied the impact of four TiO_2_ nanorods with different surface functional groups on the expression of two osteogenic differentiation hallmarks, collagen type I (COL) and osteocalcin (OCN), at both gene and protein levels. As to evidences supporting an inflammatory effect, induction of TNF-alpha and IL-6 gene expression was observed in Raw264.7 cells after 48 h incubation with 50 g/mL TiO_2_-NPs (98 ± 32 nm) [126], and an enhanced expression of pro-inflammatory genes, decreased expression of anti-inflammatory genes. Increased M1 cell surface markers (CD86, CD80, CD16/32) and decreased M2 markers (Mrc-1, Clec7a) were also found in mouse BMDMs exposed to two different TiO_2_-NPs, showing induction of a dominant pro-inflammatory activation state [143]. 

Reports on the application of a gut-on-chip system based on Caco-2 cells, in which gene expression responses upon TiO_2_-NPs exposure are evaluated and compared to a static system, suggested that the total number of differentially expressed genes and affected pathways after NPs exposure was higher under dynamic culture conditions than under static conditions [202,203,204].

### 3.3. Adverse Outcomes

From the previous sections and the information extracted from this review, a set of possible adverse outcomes following TiO_2_-NPs ingestion are proposed in Table 3.

Chronic oral exposure to TiO_2_-NPs is possibly involved in the development and progression of preneoplastic lesions in the colon. In fact, the involvement in the genesis of inflammatory bowel diseases and colorectal cancer has been recently reviewed as an AO linked to TiO_2_-NPs from the diet, showing the ability to induce a low-grade intestinal inflammation associated or not with preneoplastic lesions [205]. 

On the other hand, the demonstration of the accumulation of ingested TiO_2_-NPs in the liver [23], suggests liver injury due to oxidative stress and changes in cell signaling pathways. Long-time dietary intake of TiO_2_-NPs could induce element imbalance and organ injury in mice, and the liver displayed more serious change than other organs [206]. Risk assessment studies focused on the effects of TiO_2_-NPs ingestion, also revealed potential risk for liver, ovaries, and testes [207]. Very recently, increased serum biochemical indices, oxidative stress markers, serum cytokines, DNA fragmentation, and DNA breakages; decreased the antioxidant enzymes; and histological alterations in the liver, were also reported after TiO_2_-NPs oral administration in rats [208]. 

Reproductive toxicity has been shown in several recent publications [209], and anatase seems to be more toxic than rutile [70]. The impairment of sperm efficiency in mice following short-term TiO_2-_NPs exposure was reported [210] and TiO_2_-NPs caused pathologic changes in the mouse testis [211]. In addition, it has been proven that pregnancy exposure to TiO_2_-NPs caused delayed appearance of neurobehavioral impairments in offspring from mice when they reached adulthood [212]. Importantly, following pregnancy exposure, growth retardation and teratogenicity of TiO_2_-NPs, leading to neural tube development defects such as spinal bifida, reduction in cortical thickness, and dilatation of lateral ventricles, were reported [203]. 

Several works provide evidence of cardiac damage after TiO_2_-NPs ingestion [40,184,185] and a recent work in adult albino rats, showed alterations of histological structure of the adult rat ventricular myocardium in acute exposure [213]. Oral administration of TiO_2_-NPs to rat pups impacted also basic cardiac performance [194]. Likewise, kidney damage was suggested in by functional defects [72,145,151,184,185], together with the observation on bioaccumulation of TiO_2_-NPs in this organ [45,56,57,58,59,60,61,62,63]. The reported hematological effects point also to an adverse outcome that can be interconnected to the immune function and inflammatory response [40,185].

### 3.4. Towards an AOP Model for Ingested TiO_2_-NPs

Based on the results obtained and also on the expert judgment, a comprehensive model for a putative AOPs driven by the ingestion of TiO_2_-NPs is proposed in Figure 4 and Figure 5, considering the previously identified cellular and molecular events as the building-blocks. The literature review provides evidence on the importance of the intracellular uptake as a MIE for a cascade of KE, such as ROS generation, DNA or chromosome damage and epigenetic events. These may lead to cell cycle arrest, cell death or inflammation, potentially contributing to colorectal cancer (Figure 4). 

Conversely, the translocation of ingested TiO_2_-NPs can occur through GIT by transcytosis may also allow systemic distribution, leading to effects precluded in distal organs such as the liver. Cellular and systemic accumulation of TiO_2_-NPs occurs even when considering different exposure conditions. The evidence of the uptake of TiO_2_-NPs also verified by tissue accumulation (including cross evidence of brain and placenta barriers), may be considered as a MIE when constructing an adverse outcome landscape (Figure 5).

Three recent reports have proposed possible AOPs related to TiO_2_-NPs ingestion and further support the proposed model. An AOP for the intestine, leading to tumor formation after oral exposure to TiO_2_-NPs, has been postulated by Braakhuis and colleagues [14], and AOs such as intestinal adenomas/carcinomas were reported by Bischoff and coworkers [214]. A compilation of two AOPs leading to effects on the liver by TiO_2_, based on AOP 144 and AOP 34 of the AOP-Wiki, were reported by Brand and colleagues [15]. MIEs, such as endocytic lysosomal uptake [15]; cellular uptake in the intestine [14,214]; and alteration of gut microbiota [214] have been previously postulated. As KEs after E171 exposure, Bischoff and colleagues [214] included ROS generation, oxidative stress, persistent epithelial injury, increased cell proliferation, and DNA damage in preneoplastic lesions. Braakhuis and colleagues [16] related KEs as ROS induction, inflammation, DNA damage, and cell proliferation. Brand and colleagues [15] considered the lysosomal disruption, ROS production, mitochondrial dysfunction, cell death/injury and increased inflammatory events to be included as KEs [15]. Based on the present overview, the KEs that can be used as building blocks for an AOP in the GIT/oral exposure should clearly include: intracellular uptake, oxidative stress, cytotoxicity, inflammation, and genotoxicity-related events. Additional endpoints also showed an effect upon TiO_2_ exposure, in spite of the weight-of-evidence being lower in view of the smaller number of studies reported. 

Another AOP, the AOP 208-JAK/STAT and TGF-beta pathways activation leading to reproductive failure, has been related to TiO_2_-NPs, but is focused on the stressor UV-activated TiO_2_, not relevant for the oral route of exposure. On the other hand, interestingly, in a study from Hong, et al. (2016) [215] it was shown that the activation of the JAK/STAT pathway may be involved in the hepatic inflammation induced by chronic toxicity mice administered with a TiO_2_-NPs gavage instillation (2.5, 5, or 10 mg/kg bw) [215]. 

Overall, although there are studies demonstrating that exposure to TiO_2_-NPs may result in cytotoxicity, induction of apoptotic events, inflammation, and possibly cancer in several organs, more studies are required. Unfortunately, it is still not possible to construct a quantitative AOP driven by the ingestion of TiO_2_-NPs, because several items should be taken into consideration, such as: (a) clinical data about pre-existing diseases; (b) levels of chronic exposure, (c) size of the NPs; (d) physicochemical properties, etc.

## 4. Conclusions

The present study provides an integrative analysis of the published data on cellular and molecular mechanisms triggered after the ingestion of TiO_2_-NPs, proposing putative AOP where colorectal cancer, liver injury, reproductive toxicity, and cardiac and kidney damage, as well as hematological effects stand out as possible adverse outcomes.

The strength of evidence of this AOP proposal is based on the systematic literature review of 787 publications, by several experts in this research area, outlining biological endpoints such as cellular uptake, oxidative stress, cell death, inflammation, carcinogenicity, and other biochemical and physiological parameters, that are key events in the potential adverse effects of ingested TiO_2_-NPs. The recent transgenerational studies also point to concerns with regard to population effects of this exposure, further supporting the limitation of the use of TiO_2_-NPs in food announced by EFSA. 

The need of further studies directed to specific questions was identified, especially on: (i) chronic exposure to TiO_2_-NPs; (ii) transgenerational consequences; (iii) understanding the impact of different physicochemical characteristic of the NPs and exposure doses in the MIE, KE, and AO; and (iv) individual susceptibility and influence of previous existing health conditions on the AOs. It is also recognized that human biomonitoring studies are needed to provide relevant information on the realistic human exposure upon the widespread use of TiO_2_-NPs, allowing to link from exposure to health effects precluded in these AOPs. 

Looking beyond the scope of this review, it is proposed to use this approach to address safety concerns on other types of NPs of relevance to human health and consider these data for framing risk assessment associated with the ingestion of NPs, decreasing the level of uncertainties regarding the safe and sustainable application of emerging NPs.

Overall, AOPs at any stage of development are useful to support decision making, since they can provide a scientifically credible basis to link AOs of regulatory concern to specific pathway perturbations or biological activities, in this way guiding further toxicity testing, informing prioritization of research, and driving decision matrices. Further regulatory purposes such as risk assessment, require the ability to quantitatively define the exposure conditions under which an AO will be observed and on the degree of quantitative understanding of the relationships linking key events. In the future, quantitative AOPs can represent the bridge from descriptive knowledge to the prediction of an AO in hazard and risk assessment.

## Figures and Tables

**Figure 1 nanomaterials-12-03275-f001:**
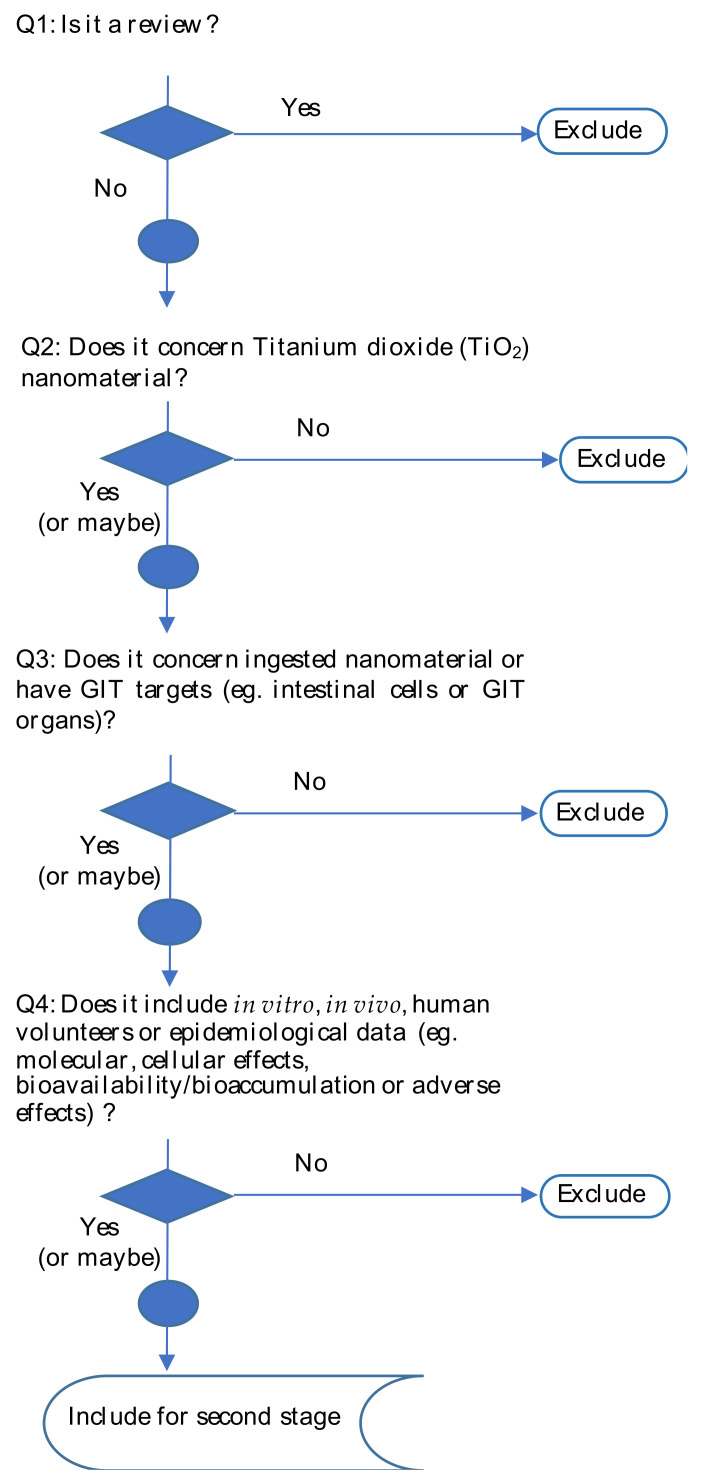
Decision tree for stage I screening.

**Figure 2 nanomaterials-12-03275-f002:**
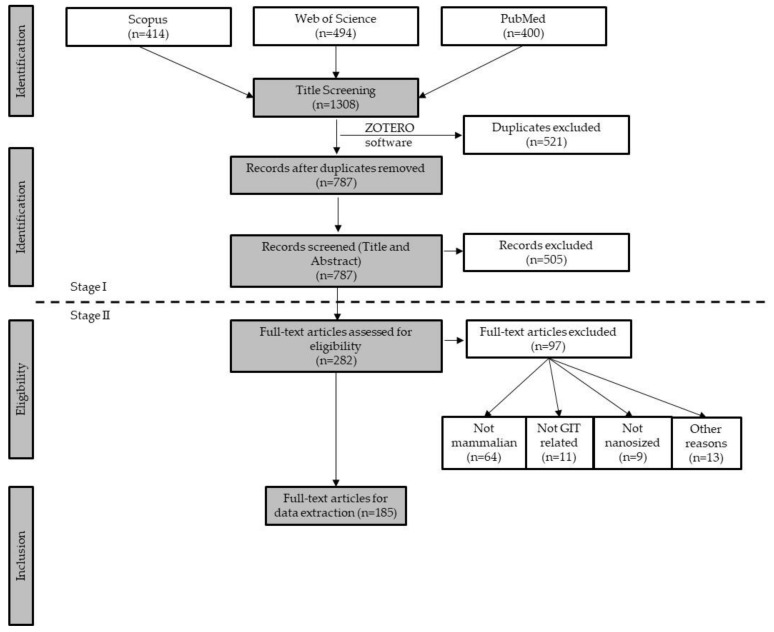
Workflow and results from Stage I and II of the literature review.

**Figure 3 nanomaterials-12-03275-f003:**
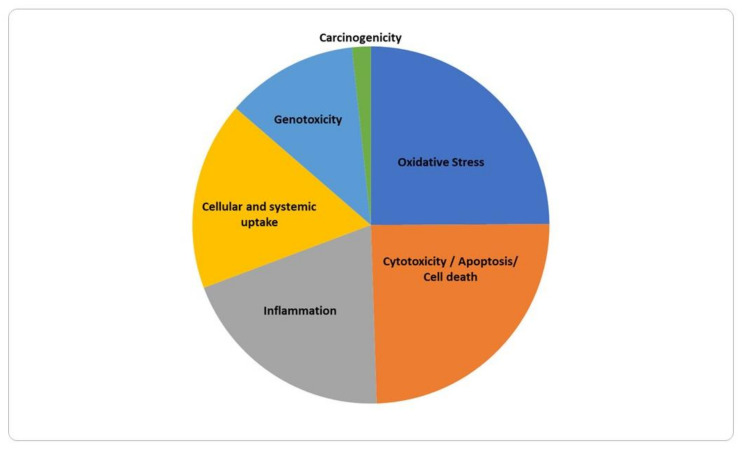
Endpoints identified in the literature that relate to TiO_2_-NPs effects.

**Figure 4 nanomaterials-12-03275-f004:**
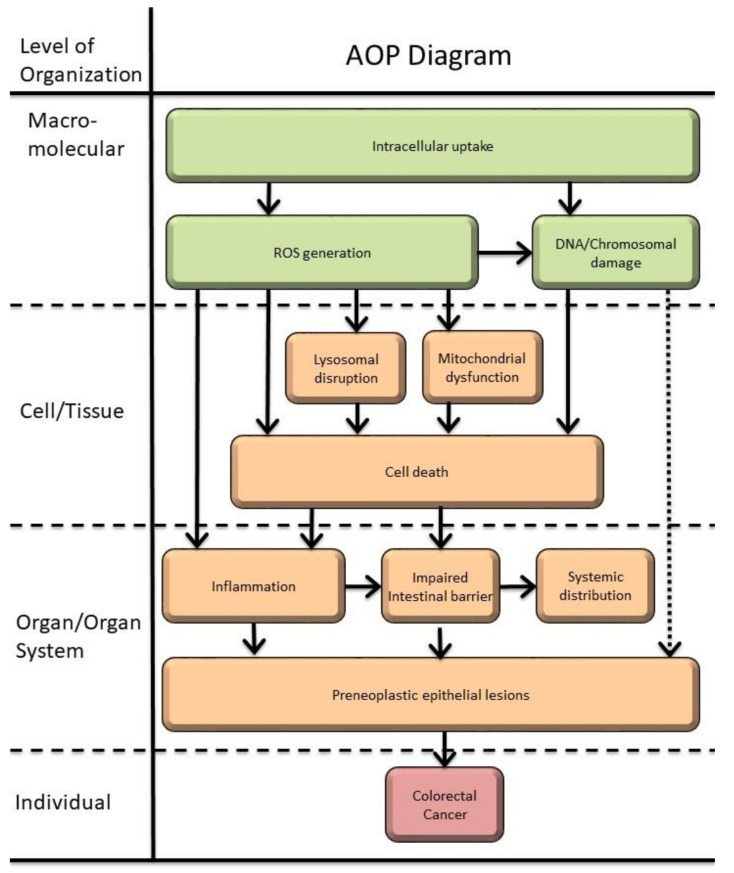
Proposal of a putative AOP model upon TiO_2_-NPs ingestion leading to colorectal cancer.

**Figure 5 nanomaterials-12-03275-f005:**
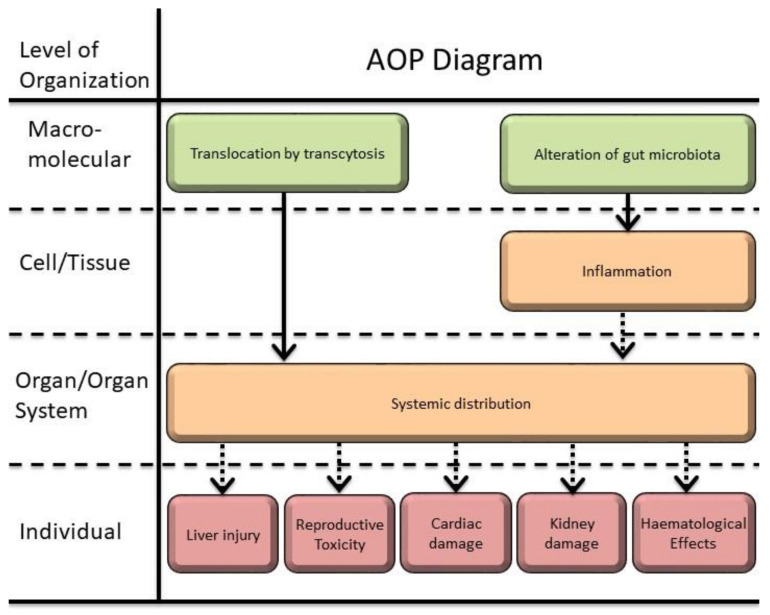
Proposal of a putative AOP model upon TiO_2_-NPs ingestion effects after systemic accumulation.

**Table 1 nanomaterials-12-03275-t001:** TiO_2_-NPs physicochemical parameters among the selected 185 studies.

TiO_2_ NM Characteristics	Categories *	No.
Crystalline phase	Anatase	81
	Mixture	39
	Rutile	23
	NA	59
Size (nm)	<25	104
	25–50	54
	50–100	45
	>100	36
	NA	23
Hydrodynamic size (DLS size, nm)	25–50	9
	>100	50
	NA	127
Specific Surface Area (SSA, m^2^/g)	<50	31
	50–100	28
	>100	20
	NA	122
Surface charge (mV)	Negative	44
	Positive	22
	NA	127

* NA, Information not available.

**Table 2 nanomaterials-12-03275-t002:** Type of studies, models used, and targets of the selected 185 papers.

Type of Study	Type of Cells/Model	Organ/Cell Target	No.
In vivo			
	Murine		
		Liver	31
		Blood	20
		Spleen	16
		Kidney	13
		Intestine	12
		Other cell types	11
	Nonmurine		
			6
In vitro			
	Human		
		GIT-related cells	38
		Other cell types	16
	Murine		15
Human Volunteers			
		GIT-related cells	3
		Other cell types	7

**Table 3 nanomaterials-12-03275-t003:** Probable adverse outcomes of ingested TiO_2_-NPs and selection of associated publications.

Adverse Outcome	Supporting References
Colorectal cancer	[7,8,19,129,178,180,181,205]
Liver injury	[23,40,56,72,78,98,119,143,184,185,206,207,208]
Reproductive toxicity	[70,114,190,191,192,193,194,195,196,197,198,199,200,201,202,203,209,210,211,212]
Cardiac damage	[40,184,185,194,213]
Kidney damage	[72,145,151,184,185]
Haematological effects	[40,56,185,187]

## Data Availability

Not applicable.

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
