# Peer review of "Adverse Outcome Pathways Associated with the Ingestion of Titanium Dioxide Nanoparticles—A Systematic Review"

_nanomaterials, 2022, doi:10.3390/nano12193275_

Round 1

Reviewer 1 Report

This paper regarding titanium dioxide nanoparticles and AOPs is highly comprehensive and well-written.  The paper captures the breadth of the literature and will make an excellent reference paper for future research papers evaluating the toxicity of these materials.  No additional edits are needed, and the paper should be accepted as written.

Author Response

Response to Reviewer 1 Comments

This paper regarding titanium dioxide nanoparticles and AOPs is highly comprehensive and well- written. The paper captures the breadth of the literature and will make an excellent reference paper for future research papers evaluating the toxicity of these materials. No additional edits are needed, and the paper should be accepted as written.

Response: The authors thank for the positive feedback on this manuscript.

Reviewer 2 Report

Topic is important and interesting. General comment is that relevance of AOP for the society could be explained. In addition more critical view of the scientific research with arguments that lead to an unequivocal conclusion about the reasonableness of using TiO2 in food could be given or at least suggested. In addition claim from the abstract that «proposing plausible AOPs which may drive policy decisions« should be better specified in regard what kind of policy authors had in mind.

Here are some other comments

1.       Abbrevation EFSA should be explained in the abstract.

2.       The majority of the papers (57.8%, 107/185) did not provide information about the dispersion method used prior to biological assays application, although the remaining studies (95/185) used sonication/ultrasonication methods. Numbers don't fit together here.

3.       Most  papers used TiO2-NPs with sizes below 100 nm, but about 20% (36/185) of the analysed papers were focused on TiO2 with more than 100 nm, and 12.4% (23/185) of the papers did not present further information about the NPs dimension. This claim should be explained in more details

Electron microscopy detector should be mentiones in regard to particles analysis.

Author Response

Response to Reviewer 2 Comments

Topic is important and interesting.

Response: Thank you for your positive feedback.

General comment is that relevance of AOP for the society could be explained.

Response: To meet this comment, a sentence was added in the introduction section (L90) and in conclusions (L1067), as follows:

“AOPs development allow to compile the existing information of the biological effects of chemicals in order to readout implications for human health and allow decision-making for risk assessors, thereby contributing to protect society from identified adverse health or ecotoxicological effects, such as cancer. “

“Overall, AOPs at any stage of development are useful to support decision making, since they can provide a scientifically credible basis to link AOs of regulatory concern to specific pathway perturbations or biological activities, in this way guiding further toxicity testing, informing prioritization of research, and driving decision matrices. Further regulatory purposes such as risk assessment, require the ability to quantitatively define the exposure conditions under which an AO will be observed and on the degree of quantitative understanding of the relationships linking key events . In the future, quantitative AOPs can represent the bridge from descriptive knowledge to the prediction of an AO in hazard and risk assessment.”

In addition more critical view of the scientific research with arguments that lead to an unequivocal conclusion about the reasonableness of using TiO2 in food could be given or at least suggested.

Response: The last sentence of the abstract was changed to clarify the reasonableness of not using TiO in food:

“Overall, the findings further support a limitation of the use of TiO2-NPs in food, as announced by European Food Safety Authority (EFSA).” (L36)

In addition claim from the abstract that «proposing plausible AOPs which may drive policy decisions« should be better specified in regard what kind of policy authors had in mind.

Response: To clarify this issue, a following sentence was added in the Introduction section (L80) and this also addressed in the conclusions (L1066):

Many regulatory agencies across the world, such as the Organization for Economic Cooperation and Development (OECD), or the European Food Safety Authority (EFSA), have recognized the potential of AOPs in supporting more efficient assessments of chemical safety, as well as for other potential stakeholders addressing for example biomedical issues or drug development.

Here are some other comments

  1. Abbrevation EFSA should be explained in the abstract.

Response: Changed accordingly.

  1. The majority of the papers (57.8%, 107/185) did not provide information about the dispersion method used prior to biological assays application, although the remaining studies (95/185) used sonication/ultrasonication methods. Numbers don't fit together here.

Response: Thank you for the detailed review. The results were checked and the sentence was reformulated as follows: “The majority of the papers (51,4%, 95/185) used sonication/ultrasonication as dispersion method used prior to biological assays application. Of notice, 42.2% (78/185) did not provide any information about dispersion methods. “ (L236).

  1. Most  papers used TiO2-NPs with sizes below 100 nm, but about 20% (36/185) of the analysed papers were focused on TiO2 with more than 100 nm, and 12.4% (23/185) of the papers did not present further information about the NPs dimension. This claim should be explained in more details.

Response: Our data extraction analysis revealed that 12% of the research do not present complementary data on the NMs’ dimension, and use them as provided by commercial sources. A sentence was added to especify this in L246.

“It should be emphasized that E171 also presents particles sized over 100 nm, and mixtures of different sizes are found in food. “(L247)

“In spite these are not in line with the recognized NM definition and recommendations to describe properties, all the papers were included for pursuing the analysis.” (L248)

Also, as argued in the main text on the following sentence: “Concerning the actual complexity of NP definition [5] it is possible that those papers could be considered outside of the definition, but we decided not to exclude them.” (L250).

Electron microscopy detector should be mentiones in regard to particles analysis.

Response: Thank you for the suggestion, the information was only presented in the Supplementary Database file. The used methodology for NPs detection was not always mentioned, and, for example, studies that used directly purchased TiO2-NPs, only presented the commercially available information. When the information was included, we decided to group all kinds of microscopy detection methods, such as TEM or SEM. An addditional sentence was included: “Electron microscopy detection was used to analyze particles in 22 studies.” (L242)

Reviewer 3 Report

This paper represent a detailed review on the potential adverse effects of TiO2 NPs present in different products. Although authors spent a lot of time in the review, I would still recommend major changes.

1. The review article is very long and  it is very hard to read. I would advise authors to reduce it as 

2. The critical question is analysis of the published data on cellular and molecular mechanisms triggered after the ingestion of TiO2-NPs and how this contribute to the further conclusion on colorectal cancer, liver injury etc. that stand out as possible adverse outcomes. This is still not clear.

3. Proposed model in Figure 5 and 6 comes out of nowhere. It is a review paper and maybe the word model is not the best. I would report more potential routes. 

4. Figure 1 and 2 can be moved to SI. 

5. In general all figures are not good quality. 

Minor:

L921 – emergent or emerging NPs?

Author Response

Response to Reviewer 3 Comments

This paper represent a detailed review on the potential adverse effects of TiO2 NPs present in different products. Although authors spent a lot of time in the review, I would still recommend major changes.

Response: Thank you for your positive feedback. English language and style spelling were checked.

  1. The review article is very long and it is very hard to read. I would advise authors to reduce it as 

Response: The manuscript text was reduced in order to enhance its legibility. Text reductions were performed in section 3.2.1, 3.2.2, 3.2.6, 3.2.7, 3.2.8 and 3.2.9, when too much detail on the same study was detected.

  1. The critical question is analysis of the published data on cellular and molecular mechanisms triggered after the ingestion of TiO2-NPs and how this contribute to the further conclusion on colorectal cancer, liver injury etc. that stand out as possible adverse outcomes. This is still not clear.

Response: This issue is addressed in section 3.4 Towards an AOP model for ingested TiO2-NPs, where the connection between the reported key events and AO is proposed. The first paragraph of this section was modified to clarify this issue (L983):

“Based on the results obtained and also on the expert judgment, a comprehensive model for a putative AOP driven by the ingestion of TiO2-NPs is proposed in Figure 5 and Figure 6, considering the previously identified cellular and molecular events as the building-blocks.”

  1. Proposed model in Figure 5 and 6 comes out of nowhere. It is a review paper and maybe the word model is not the best. I would report more potential routes. 

Response: We acknowledge the suggestion of integrating better Figure 5 and 6 in the main text, since these have been set up based on the building blocks identified in the literature search. We have improved the section text, accordingly. Also, completed the last sentence of the introduction section (L127).

For the AOP construction we have used a Powerpoint Template model, available at the OECD webpage, therefore the model seemed a suited format to use.

Given the fact that the literature search was focused only in one exposure route, oral, other potential routes of exposure would require additional literature searchs, not in the initial scope.

  1. Figure 1 and 2 can be moved to SI. 

Response: Thank you for your proposal. Analysing the information given by Figure 1, we have decided to remove it, since the text is clear about the search string content.

On the other hand, although the text description about Figure 2 was improved, we think that the decision tree is essential for the undestanding of the selection criteria on the First Stage of the literature review, thus it is important to mantained it embeded in the main text.

  1. In general all figures are not good quality. 

Response: Changed accordingly.

Minor:

L921 – emergent or emerging NPs?

Response: Changed accordingly (L1065).

Round 2

Reviewer 3 Report

authors covered all questions raised.